# Is Automated Consent in Solid GDPR-Compliant? An Approach for Obtaining Valid Consent with the Solid Protocol

Marcu Florea [1,†] and Beatriz Esteves [2,*,†]

1  Security, Technology and ePrivacy Group (STeP), University of Groningen,
   9712 GH Groningen, The Netherlands; m.o.florea@step-rug.nl
2  Ontology Engineering Group (OEG), Universidad Politécnica de Madrid, 28040 Madrid, Spain
*  Correspondence: beatriz.gesteves@upm.es
†  These authors contributed equally to this work.

**Abstract:** Personal Information Management Systems (PIMS) are acquiring a prominent role in the data economy by promoting services that help individuals to have more control over the processing of their personal data, in line with the European data protection laws. One of the highlighted solutions in this area is Solid, a new protocol that is decentralizing the storage of data, through the usage of interoperable web standards and semantic vocabularies, to empower its users to have more control over the processing of data by agents and applications. However, to fulfill this vision and gather widespread adoption, Solid needs to be aligned with the law governing the processing of personal data in Europe, the main piece of legislation being the General Data Protection Regulation (GDPR). To assist with this process, we analyze the current efforts to introduce a policy layer in the Solid ecosystem, in particular, related to the challenge of obtaining consent for processing personal data, focusing on the GDPR. Furthermore, we investigate if, in the context of using personal data for biomedical research, consent can be expressed in advance, and discuss the conditions for valid consent and how it can be obtained in this decentralized setting, namely through the matching of privacy preferences, set by the user, with requests for data and whether this can signify informed consent. Finally, we discuss the technical challenges of an implementation that caters to the previously identified legal requirements.

**Keywords:** personal information management systems; Solid; semantic web; data protection; consent

## 1. Introduction

The General Data Protection Regulation (GDPR) [1] has become the gold standard in the European Union (EU) and its effects are being globally felt in Asia, Latin America and Africa [2].

The purpose of the GDPR is twofold: on the one hand, it protects individuals in what concerns their human rights and, on the other hand, it enables the free flow of personal data (Article 1 GDPR). The EU expressed a vision that encompasses the creation of a single European market for data, where access to personal and non-personal data from across the world is secure and can be used by an ecosystem of companies, governments, and individuals to provide high-quality data-driven products and services for its citizens while ensuring that "EU law can be enforced effectively" and data subjects are still in control of what happens to their personal data [3].

In addition to the GDPR, novel data-related legislation with new data governance schemes, such as the Data Governance Act (DGA) [4], is being brought forward by the EU to build an infrastructure for data-sharing and to improve citizens' trust. In particular, trust has been proven as an important factor that positively influences the perceived usefulness

and ease of use of digital personal datastores [5] in data-handling services and allow them to share their sensitive data for the 'public good'.

However, this has not come without challenges in its interpretation and enforcement. Complex data flows where multiple parties govern the usage, storage, and collection of personal data for both distinct or shared purposes pose several challenges in elaborating privacy policies and complying with the information requirements [6]. The allocation of rights and obligations concerning the processing of personal data is structured around different roles, the main being data controllers, data processors, and data subjects. Among other obligations, data controllers (the natural or legal person who determines the purposes and means of the processing of personal data) must document and declare a lawful and transparent *purpose* to justify the processing of personal data so that the data subjects (the people whose data are processed) can make an informed decision when it comes to the usage of their personal data. Furthermore, data controllers also have an obligation to provide information to the data subjects, as according to Articles 12 [1], said information must be presented in a concise, transparent, intelligible, and easily accessible form, using clear and plain language and this is not easy to assess and implement in reality.

Compliance with transparency obligations can be dealt with by providing lengthy, complex, ungraspable privacy notices, which place a significant burden on the data subjects [7,8]. Although the necessary elements required by the law are provided, and data subjects are offered the possibility to be informed, this involves a nearly impossible exercise: to understand the myriad of terms and conditions of all the services and applications that are used these days, from smartphone applications to social media websites and personalized streaming of digital content. Thus, the information provided to data subjects fails to be an efficient tool for data subjects to monitor how their data are used or how their rights can be exercised [9].

One mode that is being largely discussed these days is *personal data sovereignty*, which presents a radical change in the paradigm of data-sharing. In this new governance model, access to data is decentralized—data subjects assume direct control over the usage and sharing of their data, a solution that promises to balance the power relationship between web users and digital platforms by promoting the development of digital services focused on users' needs [10–12].

In this context, the emergence of personal data spaces managed through Personal Information Management Systems (PIMS) is already being envisioned by the European Data Protection Supervisor (EDPS) as a mechanism to enable personal data sovereignty where "Individuals, service providers and applications would need to authenticate to access a personal storage centre" and individuals can "customize what categories of data they want to share and with whom" while keeping a track of "who has had access to their digital behaviour" and enabling data portability and interoperability [13]. Furthermore, these new user-managed systems represent the next step towards the matching of privacy terms between data subjects and data controllers and can actually play an important role in facilitating the exercise of data subjects' rights, including the rights of access, erasure, and data portability or the right to withdraw consent [14]. In this context, a set of different PIMS initiatives has been gaining prominence and adoption in the last few years, including the MyDex Community Interest Company (https://mydex.org/, accessed on 25 October 2023)—a company focused on providing portable and interoperable identity services [15], the Hub of All Things (https://www.hubofallthings.com/, accessed on 25 October 2023)—an ecosystem focused on providing contract-based access to self-sovereign data with technology implemented by Dataswyft (https://dataswyft.com/, accessed on 25 October 2023), Meeco (https://www.meeco.me/, accessed on 25 October 2023)—a company offering personal data storage and verifiable credential services [16], and the Solid project (https://solidproject.org/, accessed on 25 October 2023). Solid is a free, community-led, developer-friendly, open-source initiative that delivers on the promise of decentralizing the storage of data by relying on web standards and semantic web vocabularies to promote data and services interoperability. To fulfill this vision, the Solid specification relies on authentication

and authorization protocols to provide private, secure, and granular access to data stored in Solid's personal online datastores, the so-called "Pods" [17].

As such, there have been recent efforts to align the GDPR with personal datastores and in particular with Solid. One of the more discussed issues relies on the uncertainties generated by such decentralized systems in the definition of responsibilities under the GDPR [18,19]—while some defend that in such settings data subjects become data controllers of their own data [20], a view that clashes with existing regulations [21], others maintain that the user remains the data subject and the providers and developers of such systems are data controllers.

It is, therefore, also important to make a distinction between what can be enforced technologically and what can only be legally enforced—while technically we can restrict the data that applications can have access to, and remove the access grant when we no longer want to use them, when an app can read data from a Pod, it can also copy it, even if with Solid it does not need to do it. At this point, we enter the realm of the law—where processing must comply with several principles and rules. Although the data subject wishes, as declared by the policies that they have stored in the Pod, play an important role, their legal significance depends on how and when they are expressed [22]. Additionally, to ensure that Solid is taken up on a large scale, it is important to consider the legal implications of its development and use, the responsibilities that are associated with the roles that different actors play in this configuration, and the data subject rights that must be respected.

In addition, in order to comply with the principle of lawfulness, fairness, and transparency, data subjects must identify a ground for lawfulness as per Article 6 of the GDPR. One of the legal grounds provided under this article is the consent of the data subject. The usage of lawful grounds for processing beyond consent [19,23] or dealing with access to special categories of personal data [24] was previously discussed in the literature.

In addition to the challenges around legal bases, when it comes to the alignment of Solid with data protection requirements, a number of relevant initiatives have been materializing in recent years, mainly through academic projects and publications. Pandit analyzed this technology in terms of the actors involved, according to existing standards related to cloud technology, in order to identify GDPR issues that are still applicable in decentralized settings, such as the transparency of information, purpose limitation and exercising of data subject's rights [25]. Esposito et al. also provide a theoretical analysis of security and privacy measures to comply with the GDPR's principles of confidentiality and data minimization and to safeguard the data subjects' rights of notification, to object and to not be subjected to automated decision-making [26]. Other researchers have been focused on adding a legally compatible policy layer to Solid as a tool to express consent and determine access [27,28] and usage control [29] to data stored in Pods and on using the Verifiable Credential model to have an attribute-based access control mechanism [30]. Taking into consideration this 'law+tech' approach to the management of personal data in decentralized settings, in this work, we focus on the current efforts to introduce a policy layer to the Solid ecosystem, further developed in Section 2, as a tool to provide the necessary information and to obtain informed and valid GDPR consent. In particular, we focus on the processing of health data (considered a special category of personal data under the GDPR) for biomedical research purposes.

As such, we focus on addressing the following research question: *Can the matching between user policies and data requests, in a decentralized setting such as the Solid project, signify lawful consent under the GDPR?* The following challenges were identified for the implementation of a legally-aligned Solid ecosystem:

**Challenge 1.** Users' policies as a precursor of consent (tackled in Sections 3 and 4.1)—previous studies have shown that the current access control mechanisms supported by the Solid protocol are not enough to deal with GDPR requirements; however, there is work being developed to introduce a policy language—the Open Digital Rights Language (ODRL)—"for Expressing Consent through Granular Access Control Policies" [27]. User

policies can enable compliance with several requirements of the GDPR. Pursuant to Articles 13 and 14 GDPR, data controllers have the obligation to provide the data subject information about the processing of their personal data, and users' policies can enable communication of this information. Furthermore, information about the processing of personal data is a prerequisite for obtaining valid consent pursuant to Articles 7 and 4 (11) of the GDPR.

**Challenge 2.** Automation of consent (tackled in Section 4)—decentralized ecosystems, such as the one involving Solid Pods, rely on the existence of authorizations to provide access to (personal) data. Since the users are the ones specifying the access authorizations, said systems provide a fertile ground for research on the automation of access to resources—in this case, a data request might be automatically accepted, with no further action from the user, if the user had previously added a policy in its Pod stating that said access can be granted. Whether such automation can be considered consent under the GDPR is still up for debate. Even though there is no provision in the GDPR prohibiting the expression of consent in advance, for it to be valid, the conditions set in Article 7 and Article 4 (11) of the GDPR must also be met. In addition to the requirement of consent to be informed, the controller must be able to prove that consent was freely given, specific, and unambiguous.

**Challenge 3.** Dealing with health data for biomedical research (tackled in Section 5)—the processing of the GDPR's special categories of personal data, such as data concerning health, is prohibited by default and brings extra "burdens" to data controllers. In addition to identifying a legal basis under Article 6 GDPR, they must rely on an exception under Article 9 GDPR. There are, however, certain derogations when health data are processed for scientific research or for the management of public health (for example, Article 5 (1) (b), Article 14 (5) (b), Article 9 (2) (j), Recital 33, 52 GDPR).

To address these challenges, as the main contributions of this paper, we discuss the legal requirements that need to be addressed to build a personal information management system that aims at facilitating the expression of consent and use Solid as a use case to propose solutions for improving the current status quo of consent management. The paper is organized as follows: in Section 2, we provide an overview of Solid and relevant work in the area; in Section 3, we provide a legal overview of the distinction between providing consent and granting access to data; in Section 4, we discuss the automation of consent, in particular regarding the expression of consent in advance, the specificity of purposes, the disclosure of the identity of data controllers and the special requirements related to the usage of personal data for biomedical research; and in Section 6, we discuss future research directions and provide concluding remarks.

## 2. Background—Decentralizing the Web with Solid

In this section, we provide an overview of Solid and its main building blocks. Its access control mechanism and the existing efforts to align it with GDPR requirements are discussed in detail, including information about the ODRL profile for access control and other Solid-related work on control and privacy.

### 2.1. Solid Overview

Solid presents a radical paradigm shift in relation to today's web—by detaching data from web applications, users are given *control* over their data and *choice* over which apps they want to use with said data. This represents a major shift in power in relation to what users experience these days when they go online. By unlocking the storage of data from the hands of just a few storage providers, such as Google or Facebook, Solid gives its users the option of having a Pod—a *personal online datastore*—using their storage provider of choice or even hosting their own storage server [17]. While multiple users can use the same Solid server to host their data Pod, Solid's ultimate goal is to give its users the highest possible degree of decentralization—one Pod per person, or even multiple Pods per person, with a granular access control mechanism where they can choose which people and apps have access to their Pod, to a particular container of resources stored in their Pod or even to an individual Pod resource. In this scenario, applications act as clients that can read

and/or write data from/to different Pods, without storing it in their own servers. Therefore, beyond giving people control over their data, such an ecosystem "fosters innovation and competition through separate markets for data and applications" [22].

Solid's two main building blocks (https://solidproject.org/TR/, accessed on 25 October 2023) are its authentication (https://solidproject.org/TR/oidc, accessed on 25 October 2023) and authorization protocols (https://solid.github.io/authorization-panel/authorization-ucr/, accessed on 25 October 2023). The authentication protocol is related to the identification of agents—the WebID specification (https://solid.github.io/webid-profile/, accessed on 25 October 2023) is used to identify agents through URLs, which when dereferenced, direct to a profile document that can contain information describing the agent it identifies. The authorization protocol deals with the server's responses to requests of particular agents, in other words, it is the access control mechanism of Solid. Furthermore, the current version of the Solid protocol specification (https://solidproject.org/TR/protocol, accessed on 25 October 2023) states that, for a Solid server to be compliant, it "MUST conform to either or both Web Access Control (WAC) and Access Control Policy (ACP) specifications" (https://solidproject.org/TR/wac and https://solidproject.org/TR/acp, respectively, accessed on 25 October 2023). Further details on the authorization protocol will be given in Section 2.2. A third building block is now being developed—the Solid Application Interoperability specification (https://solid.github.io/data-interoperability-panel/specification/, accessed on 25 October 2023). Said specification details how agents and applications can interoperate and reuse data from different sources.

### 2.2. Access Control in Solid

As pointed out in the previous section, access control in Solid can currently be determined with two different specifications, WAC and ACP. While the Solid protocol mandates that the servers where the Pods are hosted conform to only one of the WAC or ACP access authorizations, Solid applications must comply with both or else they take the risk of not being usable by half of the ecosystem. Both solutions rely on IRIs to identify resources and agents, while WAC uses Access Control Lists (ACLs) to store authorizations, defined per resource or inherited from the parent resources, and ACP uses Access Control Resources (ACRs) to describe who is allowed or denied access to resources and access grants to represent already authorized accesses.

As illustrated by Listings 1 and 2, neither WAC nor ACP have the coverage to model the GDPR's information requirements (Articles 13 and 14 [1]) for the processing of personal data, in particular when it comes to the modeling of the purpose for processing, personal data categories, legal basis or even information on the identity of the data requester. To overcome this issue, research has been developed in the area of using privacy policy languages as a resource to represent information related to the GDPR's transparency requirements [31] and for the particular context of healthcare data as well [32]. Moreover, Esteves and Rodríguez-Doncel [31] describe the ODRL model as a mature solution that is "ready to be used for representing privacy-related rights and obligations" as it is open access, has good documentation, is a W3C standard for digital rights expression and has an extension mechanism that can be easily used to implement an ODRL profile for the Solid ecosystem.

**Listing 1.** WAC authorization that makes a WebID profile readable by any agent.

```
<#public> a acl:Authorization ;
    acl:agentClass foaf:Agent ;
        acl:accessTo <https://solidweb.me/besteves4/profile/card> ;
        acl:mode acl:Read .
```

**Listing 2.** ACP authorization that makes a WebID profile readable by any agent using any client application.

```
PREFIX acp: <http://www.w3.org/ns/solid/acp#>
PREFIX acl: <http://www.w3.org/ns/auth/acl#>

<#public> acp:grant acl:Read ;
    acp:context [
    acp:agent acp:PublicAgent ;
        acp:target <https://solidweb.me/besteves4/profile/card> ;
            acp:client acp:PublicClient ;
            acp:issuer <https://solidweb.me/>
    ] .
```

ODRL (http://www.w3.org/ns/odrl/2/, accessed on 25 October 2023) [33] is a W3C standard for policy expression that includes an information model and a vocabulary of terms. It provides a convenient extension mechanism, through the definition of ODRL profiles (https://www.w3.org/profile-bp/, accessed on 25 October 2023), that can be used to create policies for different use cases, from software licenses to access and usage control policies. Since ODRL is not domain specific, i.e., it can be extended to create policies for financial (https://w3c.github.io/odrl/profile-temporal/, accessed on 25 October 2023) or language (https://rdflicense.linkeddata.es/profile.html, accessed on 25 October 2023) resources, it means that it is also not equipped to deal with legal requirements. To this end, the ODRL profile for Access Control (OAC) (https://w3id.org/oac, accessed on 25 October 2023) makes use of ODRL's deontic representation capabilities and connects them with the Data Privacy Vocabulary (DPV) (https://w3id.org/dpv, accessed on 25 October 2023) [34] to invoke data protection-specific terms. DPV provides an ample set of taxonomies that can be used to specify entities, legal basis, personal data categories, processing activities, purposes, or technical and organizational measures. Therefore, by integrating the usage of ODRL and DPV, OAC allows Solid users to express their privacy preferences and requirements over particular types of data, purposes, recipients, or processing operations at distinct levels of specificity—from broad, e.g., allow data use for scientific research, to narrow policies, e.g., prohibit sharing a particular resource with a particular application. Figure 1 presents a diagram with the main concepts defined in OAC to express such policies. Requests for access, either from other users or from applications or services, can be modeled in a similar manner and stored in the Pod to have a record of said requests. Listings 3 and 4 illustrate an example of a user policy as an `odrl:Offer` and an example of a data request as an `odrl:Request`, respectively.

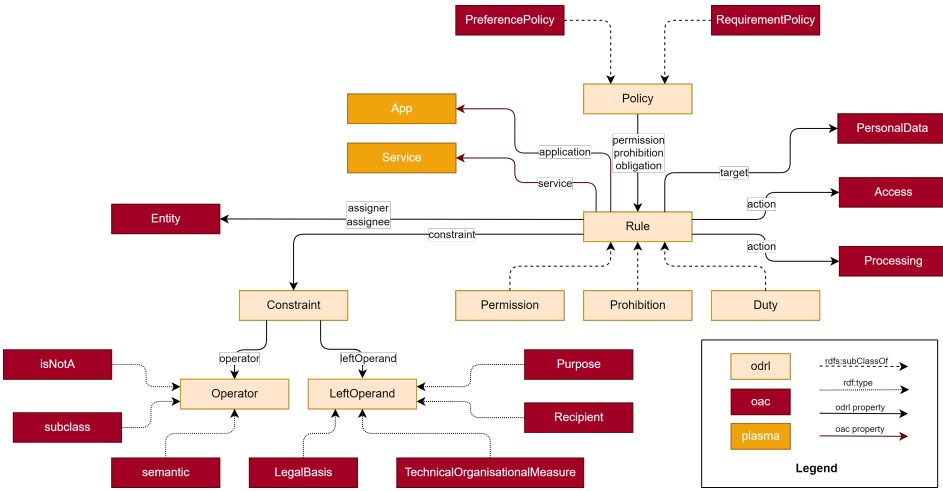

**Figure 1.** Core concepts of the ODRL profile for Access Control (OAC).

**Listing 3.** An example ODRL offer policy generated by https://solidweb.me/besteves4/profile/card#me, stating that health records data can be accessed for the purpose of health, medical or biomedical research.

```
PREFIX odrl: <http://www.w3.org/ns/odrl/2/>
PREFIX dcterms: <http://purl.org/dc/terms/>
PREFIX xsd: <http://www.w3.org/2001/XMLSchema#>
PREFIX oac: <https://w3id.org/oac#>
PREFIX dpv: <https://w3id.org/dpv#>
PREFIX duodrl: <https://w3id.org/duodrl#>
PREFIX ex: <https://example.com/>

<https://example.com/offer1> a odrl:Offer ;
    dcterms:description "Offer to read health records data for health, medical or biomedical research." ;
    dcterms:creator <https://solidweb.me/besteves4/profile/card#me> ;
    dcterms:issued "2023-05-30T17:26:35"^^xsd:dateTime ;
    odrl:uid ex:offer1 ;
    odrl:profile oac: ;
    odrl:permission [
        dpv:hasContext dpv:Optional ;
        odrl:assigner <https://solidweb.me/besteves4/profile/card#me> ;
        odrl:target oac:HealthRecord ;
        odrl:action oac:Read ;
        odrl:constraint [
            dcterms:title "Purpose for access is to conduct health, medical or biomedical (HMB) research." ;
            odrl:leftOperand oac:Purpose ;
            odrl:operator odrl:isA ;
            odrl:rightOperand duodrl:HMB ] ] .
```

By integrating the usage of such a policy layer in the Solid ecosystem, the matching of users' preferences and requests for data is possible and can be automated. OAC's proposed matching algorithm consists of checking for subsumption between data requests and user policies—if the data request satisfies the users' policies, then access can be provided to the Pod. On the other hand, if any prohibitions are found in the users' policies that match the data request, access to the Pod is denied. The result of the matching is stored in the Pod for record keeping and future inspection. Thus, OAC will be used as our motivating scenario. While the decision to deny access based on user policies can be interpreted as the exercise of a data subject right, e.g., the right to object in Article 21 GDPR, this article focuses on whether the positive result of the matching can signify consent.

### 2.3. Other Related Works

The issue of control and privacy in Solid has been further explored by academia and industry. Beyond access, research on usage control has also been developed [29,35], with the main goal of creating tools to enforce policies and ensure that data are being used according to the users' preferences after access has been provided. In addition, the exercising of the GDPR's data subject rights, in particular of the Right to Data Portability [36] and the Right of Access [37], has been proven to be facilitated through the usage of Solid. Digita, a Belgium-based startup commercializing Solid solutions (https://www.digita.ai/, accessed on 25 October 2023), also published a research report reflecting on the applicability of the GDPR's requirements to Solid implementations, in particular, regarding data exchange with consent [23]. Recent efforts also promoted a tool to generate and store OAC policies in Solid Pods [38] and evaluated the usage of the Solid Application Interoperability specification to create a User Interface for users to evaluate data requests [39]. Hochstenbach et al. are developing RDF Surfaces (https://w3c-cg.github.io/rdfsurfaces/, accessed on 25 October 2023), a Notation3 language that intends to bring first-order logic to the Semantic Web and therefore can be used to "provide enforcement of data policies using logic-based rules" [40].

**Listing 4.** An example ODRL Request policy made by https://solidweb.me/arya/profile/card#me, using the https://example.com/healthApp application, to use health records data from https://solidweb.me/besteves4/profile/card#me to conduct research on arterial hypertension disease.

```
PREFIX odrl: <http://www.w3.org/ns/odrl/2/>
PREFIX dcterms: <http://purl.org/dc/terms/>
PREFIX rdfs: <http://www.w3.org/2000/01/rdf-schema#>
PREFIX xsd: <http://www.w3.org/2001/XMLSchema#>
PREFIX oac: <https://w3id.org/oac#>
PREFIX duodrl: <https://w3id.org/duodrl#>
PREFIX dpv: <https://w3id.org/dpv#>
PREFIX ex: <https://example.com/>

<https://example.com/request1> a odrl:Request;
    dcterms:description "Request to use health records data for research on arterial hypertension." ;
    dcterms:creator <https://solidweb.me/arya/profile/card#me> ;
    dcterms:issued "2023-05-31T18:15:56"^^xsd:dateTime ;
    odrl:uid <https://example.com/request1> ;
    odrl:profile oac: ;
    odrl:permission [
        odrl:assignee <https://solidweb.me/besteves4/profile/card#me> ;
        odrl:assigner <https://solidweb.me/arya/profile/card#me> ;
        oac:application <https://example.com/healthApp> ;
        odrl:action oac:Use ;
        odrl:target oac:HealthRecord ;
        odrl:constraint [
            dcterms:title "Purpose for access is to conduct research on arterial hypertension." ;
            odrl:leftOperand oac:Purpose ;
            odrl:operator odrl:eq ;
            odrl:rightOperand ex:HypertensionResearch ] ] .

ex:HypertensionResearch a dpv:Purpose ;
    rdfs:subclassOf duodrl:HMB ;
    rdfs:label "Conduct research on arterial hypertension disease." .
```

In the particular field of health research, a Solid-powered platform has been developed to manage data requests and provide consent for health-related research using DPV [41]. Solid is also being tested by the United Kingdom's (UK's) National Health Service (NHS) to collect and process patient data from several systems, which is then hosted in individual patient Pods owned by the patients, who can authorize their healthcare professionals to have access to the data [42].

## 3. Describing the Distinction between Consent and Granting Access to a Resource

This section focuses on the distinction between the legal notion of consent and the technical means for authorizing access and use of a resource stored in a Solid Pod.

### 3.1. Consent vs. Other Grounds for Lawfulness

The "default setting" of a Solid Pod is "closed" as, in the absence of an authorization, the resources in a Pod cannot be accessed by Solid applications. Authorizations can be granted directly by the user (approving requests from individual applications, as they are received) or indirectly (by setting rules on access in advance).

From the perspective of the GDPR, the authorization of the user for the processing of personal data is sometimes *unnecessary* and other times *not sufficient* for Solid applications to process personal data in a lawful manner. It can be *unnecessary* since apart from consent, there are, pursuant to Article 6 (1) of the GDPR, five grounds that do not involve the active choice of the data subject [43]. For example, if the data subject entered into a contract with the entity that requests access (the data controller) and the processing of personal data is necessary for the performance of that contract (Article 6 (1) (b) GDPR), there is no need for

the consent of the data subject [44]. Similarly, if the processing of personal data is necessary for the purposes of the legitimate interest of the data subject or for compliance with legal obligations, there is no need for the authorization of the data subject (Article 6 (1) (f) GDPR).

Data subject authorization is *not sufficient* if the processing is based on Article 5 (1) (a) GDPR—the consent of the data subject—and the conditions for obtaining valid consent are not observed. In order for it to be valid, consent must be freely given, specific, informed, and unambiguous (Article 4 (11) GDPR). An authorization to "read" as per the WAC and ACP authorization mechanisms (Listings 1 and 2) will most likely fail to comply with the requirements of consent.

By juxtaposing the legal requirements and the technical features of Solid, we can observe that there are two clusters of problematic cases: (i) instances when app providers comply with the requirements of Article 5 (1) (b)–(f) GDPR and respect the principle of lawfulness, but do not have access to the personal data, because no authorization was granted by the user and (ii) instances when the app providers rely on consent (Article 5 (1) (a) GDPR), as a ground for lawfulness, but the authorization does not comply with the requirements of obtaining valid consent.

While the first cluster of cases is very relevant to the functioning of Solid, this paper will focus on the second one, exploring the legal requirements for obtaining valid consent and the technical means available to this end.

### 3.2. OAC: From Notice to Automated Consent

Apart from the requirements for lawfulness, the GDPR provides information obligations that affirm the principle of transparency, one component of Article 5 (1) (a) of the GDPR. Articles 12 to 14 of the GDPR require data controllers to take appropriate measures to provide the data subject with information regarding the processing of their personal data, irrespective of the ground of lawfulness chosen. Therefore, even when data subjects are not required to authorize the processing of their personal data, data controllers have an obligation to provide information regarding the processing of the personal data.

Recital 59 GDPR provides that modalities should be provided for facilitating the exercise of the data subject's rights. Users' policies can function as an information mechanism, enabling data subjects to easily understand the details surrounding the processing of their personal data.

Moreover, the result of the matching exercise is stored in a known location in the Pod, where the data subjects can access and check the result, enabling them to easily comprehend whether the agreed specific conditions for processing personal data differ from the pre-set of preferences stored in the Pod. Listing 5 presents the ODRL agreement generated as the result of the matching exercise between the data request in Listing 4 and the user policy in Listing 3. Since the personal data type in the user policy, i.e., health records, matches the requested personal data type, and the requested purpose for processing is "to conduct research on arterial hypertension", which is a subclass of health, medical or biomedical research—the purpose allowed by the user policy-—an agreement between the data subject and controller is generated and stored in the Pod, allowing the controller to access said data. Such a record also allows the user to check which entities requested access to the data.

The list of elements that has to be made available to the data subject is mentioned in Articles 13 and 14 GDPR. Currently, OAC focuses on the types of data, legal basis, purpose and data controllers accessing the information, leaving out several elements mentioned in Article 13 (1) of the GDPR: the contact details of the controller and the controller's representative, the contact details of the data protection officer, the legitimate interest pursued by the controller or a third party (when the processing is based on Article 6 (1) (f) GDPR), the recipients or categories of recipients, and information about data transfers to third countries or international organizations. In addition, OAC does not include the requirements of Article 13 (2) GDPR: the period for which their data will be stored, the existence of data subject rights, if the provision of data is a requirement to enter into a

contract or a statutory obligation, the consequences of failure to provide the information and the existence of automated decision-making.

**Listing 5.** An example ODRL Agreement policy that grants https://solidweb.me/arya/profile/card#me read access to health records data from https://solidweb.me/besteves4/profile/card#me to conduct research on arterial hypertension disease.

```
PREFIX odrl: <http://www.w3.org/ns/odrl/2/>
PREFIX dcterms: <http://purl.org/dc/terms/>
PREFIX xsd: <http://www.w3.org/2001/XMLSchema#>
PREFIX oac: <https://w3id.org/oac#>
PREFIX dpv: <https://w3id.org/dpv#>
PREFIX duodrl: <https://w3id.org/duodrl#>
PREFIX ex: <https://example.com/>

<https://example.com/agreement1> a odrl:Agreement;
    dcterms:description "Agreement to read health records data for research on arterial hypertension." ;
    dcterms:issued "2023-05-31T18:20:06"^^xsd:dateTime ;
    odrl:uid <https://example.com/agreement1> ;
    odrl:profile oac: ;
    dpv:hasDataSubject <https://solidweb.me/besteves4/profile/card#me> ;
    dpv:hasDataController <https://solidweb.me/arya/profile/card#me> ;
    dpv:hasLegalBasis dpv:Consent ;
    dcterms:references ex:offer1, ex:request1 ;
    odrl:permission [
        odrl:assignee <https://solidweb.me/arya/profile/card#me> ;
        odrl:assigner <https://solidweb.me/besteves4/profile/card#me> ;
        oac:application <https://example.com/healthApp> ;
        odrl:action oac:Read ;
        odrl:target oac:HealthRecord ;
        odrl:constraint [
            dcterms:title "Purpose for access is to conduct research on arterial hypertension." ;
            odrl:leftOperand oac:Purpose ;
            odrl:operator odrl:eq ;
            odrl:rightOperand ex:HypertensionResearch ] ] .
```

In the absence of these elements or if there are differences between them and the request for accessing the data, the data subject should be provided with additional information, specific to the data request. As previously explained in Section 2.2, access to resources in Solid depends on authentication and authorization protocols. Access can be granted by data subjects when they start using a new application, through an authorization dialogue, such as the example provided in Figure 2 (https://communitysolidserver.github.io/CommunitySolidServer/7.x/, accessed on 25 October 2023), or can be pre-set in the Pod in advance, such as the example provided in Figure 3 (https://docs.inrupt.com/user-interface/podbrowser/, accessed on 25 October 2023).

There are two other challenges regarding the information mechanism that should be considered. The first is related to the manner in which the information is presented to the data subject. Pursuant to Article 12 of the GDPR, data controllers have an obligation to provide the information in a concise, transparent, intelligible, and easily accessible form, using clear and plain language. While the details of the matching are made available in a known location of the Pod, it might be necessary to implement an interface that presents the result of the matching, especially the new, dynamic elements, to the data subject, ensuring that they have read and understood this information.

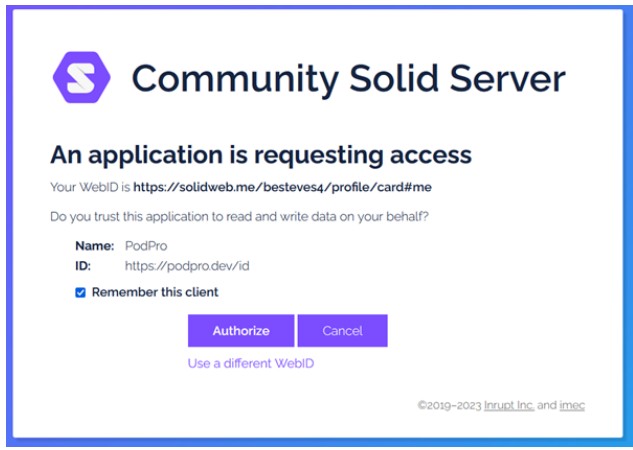

**Figure 2.** Screenshot of the authorization dialogue of the Community Solid Server (CSS) Pod provider.

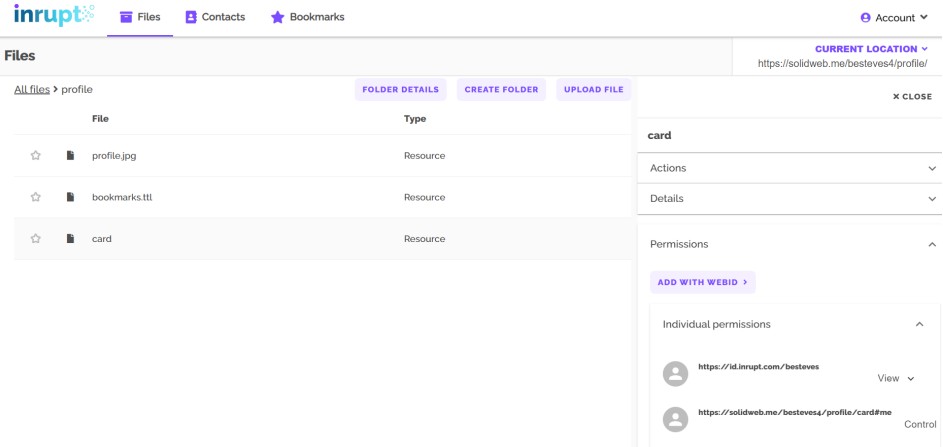

**Figure 3.** Screenshot of Inrupt's PodBrowser app to manage data and access grants.

The second challenge refers to the timing of the notification. Articles 13 and 14 of the GDPR set different rules depending on whether the data are collected directly from the data subject or from another entity. The rules do not refer to data intermediary services, leaving a gap for interpretation. If the Pod provider is considered a data controller [25], the personal data are not obtained directly from the data subject. According to Article 14 of the GDPR, in this case, the information must be provided at the time when the data controller communicates with the data subject for the first time. The request for access to the Pod can be considered a communication with the data subject, placing the information requirement at the moment when the request is filed. If the Pod provider is not considered a data controller, but merely a software tool used by the data subject, the data are obtained directly from the data subject and the information must be provided at the time when it is obtained. Although there are different rules depending on the entity disclosing the data, in both interpretations, the information must be presented the moment when the request reaches the Pod, at the latest.

Both options—access provided at the time of starting to use the app or pre-set authorizations—can take various forms, depending on the technical access control mechanism that is implemented in the server where the Pod is hosted, as the servers are not obliged to implement both authorization protocols (WAC and ACP) promoted by Solid, as was previously discussed in Section 2.2, and on the interfaces used to interact and manage the data and the access grants stored in the Pod.

The outcome of the matching exercise is a mapping between the user's policies and the specifics of the request for accessing the data, emphasizing the differences between the two. This process can lower the burden of data subjects in reading and comprehending the information related to the processing of their personal data.

Moreover, the informed character of consent is merely one element necessary for obtaining valid consent. After the individuals are informed, they must indicate their wishes by a statement or by a clear affirmative action, signifying agreement to the processing of their personal data. Pursuant to Article 4 (11) GDPR, consent must be freely given, specific, informed and unambiguous and these requirements are further developed in the European Data Protection Board (EDPB) and Article 29 Data Protection Working Party (WP 29) guidelines [45–47].

Consent must be granular, the purposes of processing cannot be bundled, and the data subject must be informed of the consequences of refusal. A mere acknowledgment or an agreement for sharing a resource does not necessarily signify consent.

## 4. Can Consent be Automated?

In order to determine whether automation is possible, it is necessary to look into the rationale behind the requirement to obtain consent. According to Jarovski, consent is important because it preserves autonomy and enables data subjects to have agency regarding the use of their personal data. The data subjects must (i) understand the conditions and risks regarding the processing of their personal data, (ii) decide among existing options, and (iii) express their choice, while being aware of the possibility of changing it in the future [48]. In the absence of protective measures, there is a risk that consent becomes meaningless. Solove described the shortcomings of privacy self-management separating between cognitive limitations, related to people's ability to make decisions (uninformed individuals, skewed decision-making) and structural limitations that hamper the adequate assessment of the costs and benefits of consenting to various forms of processing of personal data (the problem of scale, the problem of aggregation and the problem of assessing harm) [49]. Jarovski distinguishes between problems related to cognition (complexity, manipulation and behavioral biases), and information overload (length, uniquity, insufficient information, lack of intervenability, and lack of free choice) [48]. Presenting individuals with the result of the matching and asking for their consent each time a new request for access to personal data reaches the Pod might not scale well [50].

One solution to these problems is to automate consent [48,51]. This option has been criticized [51] because of the complexity of the decisions regarding the processing of personal data. Consent involves weighing the risks and benefits of processing. This is a contextual assessment that takes into account many variables, including the likelihood and gravity of harm, making it difficult to imagine how an automated system can weigh all the arguments for and against the processing of personal data.

The question is whether matching preferences configured in OAC in advance is sufficient to signify a choice that complies with the legal requirements for expressing valid consent. Consent is generally viewed as binary: an individual expresses her/his wishes representing agreement or disagreement with the processing of her/his personal data. The standard model of consenting involves the following steps: the data controller sends a request for consent and the data subject accepts or rejects it. Pre-setting access permissions in the Pod in advance switches this order. The data subjects make the first move and set their preferences concerning the processing of personal data in advance. Even though the GDPR does not provide a framework for the interaction between the data subject and a technology-based system in expressing consent, the possibility of expressing consent by technical settings is suggested in Recital 32 GDPR: "Consent should be given by a clear affirmative act establishing a freely given, specific, informed and unambiguous indication of the data subject's agreement [. . . ]. This could include [. . . ] *choosing technical settings* for information society services or another statement or conduct which *clearly indicates in this context the data subject's acceptance* of the proposed processing of his or her personal data [. . . ]".

As explained in [27], there are two possible levels of automation and both start with a request for accessing personal data. However, what follows after this is different. In the *first option*, the result of the matching is presented to the data subject, and the data

subject is requested to provide her/his consent. In this case, the matching process enables information and helps the data subject to make an informed choice, but consent is only expressed after the moment when the app provider/developer requests permission to process the personal data, i.e., when the user wishes to use the app. In the *second option*, access to personal data is granted automatically, based on the preferences of the data subject, which are expressed in advance. The first type of automation affirms the principle of transparency and helps the data subject comprehend the information in the privacy policy and apply it to the choice that she/he is requested to make. This partially solves the cognitive problems that data subjects are facing. This would make it easier for the data subject to be informed, but would still require them to express a choice repeatedly, which will be a problem in scaling the model. If a new consent action is requested for each instance of access, this might overwhelm the data subject with too many requests. The second option grants automated access to entities that request permission to access resources in a Pod, if the preferences of the individual user match the request for consent without the requirement for a new authorization.

*4.1. Expressing Consent in Advance*

There is no provision in the text of the GDPR that forbids the expression of consent in advance and Recital 32 GDPR hints in this direction. However, in order to be valid, consent must refer to specific circumstances that may arise in the future [52]. This and the following sections explore whether OAC captures the elements required by law to express valid consent.

The European Court of Justice held in Orange Romania stated that, before agreeing to the processing of their personal data, the data subject should have obtained information "relating to all the circumstances surrounding that processing" and that the data subject should be informed of the consequences of their consent [53]. The idea of automated implementation of user preferences was heavily discussed in the United States. The Do Not Track (DNT) Initiative (https://www.eff.org/issues/do-not-track, accessed on 25 October 2023) and the Platform for Privacy Preferences (P3P) (https://www.w3.org/P3P/, accessed on 25 October 2023) are two examples in this respect. Neither succeeded in being taken up on a large scale, but both can constitute helpful examples in developing a system based on consent for Solid. While the DNT Initiative is focused on behavioral-based advertising and is more connected to the right to object [54], the P3P initiative resembles in various aspects the Solid authorization mechanism as it allowed webpages to "express their privacy practices in a standard format that can be retrieved automatically and interpreted easily by user agents" and "enable an expanded ecosystem in which web sites would consistently inform web user agents of personal data collection intentions and web users would configure their individual user agents to accept some practices automatically [...]" [55]. One of the issues discussed in relation to P3P was the consistency between the human-readable privacy notices presented to individuals and the statements made in formal language with the P3P standard [56]. This can also be a challenge in Solid, in case the terms and conditions and privacy policies of the applications differ from the formalization that is implemented. WP 29 also commented on the P3P initiative [57]. One critique was that the platform could mislead controllers, making them believe that they were discharged of certain obligations if the data subjects agreed to the processing of their personal data. These issues can also be relevant in the development and implementation of Solid.

However, Solid is different from P3P, because it provides users with a decentralized storage solution for their personal data, with a permission-based access control mechanism, i.e., only with the presence of an authorization stored in the Solid Pod can an application or user access data stored in that same Pod. In addition, in a decentralized environment such as Solid, there is no need to transfer data, as access can be provided to any web user or service, removing the need for services to store or keep copies of the users' data, through Solid's authorization and authentication mechanisms. These mechanisms can also be used to keep records of access to and usage of data, which then can be used to check whether

web services are doing what they state in their policies and nothing more—one of the main issues that contributed to the failure of P3P was the lack of enforcement of web services policies (as cited in the P3P specification "no enforcement action followed when a site's policy expressed in P3P failed to reflect their actual privacy practices"). As such, these mechanisms also allow users to have more transparency and control over how their data are being used, which comes as an improvement over P3P's lack of consistency between human and machine-readable data handling policies that did not have such controls. In addition, as previously stated in Section 2.3, there is research being conducted on the implementation of usage control mechanisms for Solid.

In a rather recent development, the Commissioner for Justice and Consumers, Didier Reynders, launched a reflection on how better to empower consumers to make effective choices regarding tracking-based advertising models (https://commission.europa.eu/live-work-travel-eu/consumer-rights-and-complaints/enforcement-consumer-protection/cookie-pledge_en, accessed on 25 October 2023). The problem that it tries to solve is similar to the difficulties around accessing content in Solid Pods.

The possibility of pre-configured choices is also mentioned in the context of the use of cookies in Recital 66 of Directive 2009/136/EC [58], modifying the ePrivacy Directive [59] and in national laws implementing the ePrivacy Directive, which refers to the storage of information or gaining access to information stored in the terminal equipment of a subscriber or users of publicly available electronic communications services. Some national laws implemented the text by allowing the expression of consent via technical means. For example, the implementing law in Romania [60] refers to "using the settings of the Internet browsing application or other similar technologies" for expressing consent, while in Finland, the ombudsman and the Transport and Communications agency expressed different views on the validity of consent related to cookies given through browser settings (https://cookieinformation.com/resources/blog/finland-changes-cookie-rules/, https://tietosuoja.fi/-/apulaistietosuojavaltuutettu-maarasi-yrityksen-muuttamaan-tapaa-jolla-se-pyytaa-suostumusta-evasteiden-kayttoon, accessed on 25 October 2023). On a technical level, several initiatives of *'privacy signals'* have been emerging as a tool for users to communicate their preferences, e.g., DNT [54], the Global Privacy Control (GPC) [61], or the Advanced Data Protection Control (ADPC) [62] efforts. However, despite their benefits, they are still lacking adoption, are not standardized for interoperability nor legally tested to fulfill ePrivacy requirements [63].

The next sections will discuss some of the building blocks necessary for expressing consent in advance and debate how Solid can be adapted to comply with these requirements.

### *4.2. Expressing Specific Consent*

To determine whether consent can be automated, it is relevant to discuss the level of granularity required by the law and whether and how this can be achieved in a technical implementation.

Pursuant to Article 4 (11) of the GDPR, consent must be a specific indication of the data subject's wishes that signifies agreement to the processing of personal data. The expression "indication of wishes" is rather indeterminate. One might question if the wishes of the data subject refer to the categories of data, the purpose of processing, the processing operations, the identity of the data controller(s), or a correlation between them. The European Court of Justice mentions in Case C-61/19 Orange Romania [53] para 38 "'specific' indication of the data subject's wishes in the sense that it must relate specifically to the processing of the data in question and cannot be inferred from an indication of the data subject's wishes for other purposes".

The EDPB mentions in its guidance that the requirement of specificity is related to user control and transparency [45] and outlines three requirements:

a. Purpose specification as a safeguard against function creep;
b. Granularity of consent requests;

c. Clear separation of information related to obtaining consent for the data processing activities from information about other matters.

According to Eleni Kosta, specificity is respected when the relation between the personal data and the processing for which the data subject wishes to consent, are clearly defined and the conditions surrounding the processing are explained. The consent statement should be so specific that it safeguards the right to informational self-determination [52].

Recital 42 GDPR provides that consent is informed if the data subjects are aware at least of the *identity of the controller* and the *purposes of the processing* for which the personal data are intended. However, the text does not refer to the level of detail required for their description.

### 4.2.1. Specificity of Purpose and Processing Operations

The text of the GDPR refers to the specificity of purpose, but also to the specificity of the processing operations. Article 6 (1) (a) GDPR pinpoints the specificity requirement in relation to the *purpose* of processing: "Processing shall be lawful only if and to the extent that [. . . ] the data subject has given consent to the processing of his or her personal data for one or more specific purposes." However, the preamble of the GDPR suggests that consent is expressed not only for separate purposes but also for separate data processing operations. Recital 43 GDPR provides that "Consent is presumed not to be freely given if it does not allow separate consent to be given to different personal data processing operations despite it being appropriate in the individual case". There is a categorical difference between purposes and processing operations. While purposes describe the reason or the objective for processing personal data, processing operations refer to the actions performed in relation to the personal data. Article 4 (2) GDPR provides several examples of operations: "collection, recording, organization, structuring, storage, adaptation or alteration, retrieval, consultation, use, disclosure by transmission, dissemination or otherwise making available, alignment or combination, restriction, erasure or destruction". Several operations might be necessary to reach a purpose. Several purposes can be reached by the same processing operation (e.g., the use of data, which has a very broad scope).

One possible explanation lies in Recital 32 GDPR, which refers to the relation between purposes and processing operations: "Consent should cover all processing activities carried out for the same purpose or purposes. When the processing has multiple purposes, consent should be given for all of them". The paragraph can be constructed as requiring a connection between a specific purpose and the processing activities necessary to reach that purpose. If a processing activity covers more than one purpose, consent must be provided for all of the purposes. Conversely, if a purpose is reached by multiple processing operations, consent must be provided for all of them.

The WP 29 provides in its opinion on consent that *specific* consent is intrinsically linked to the fact that consent must be informed [46]. There is a requirement for granularity of consent with regard to the different elements that constitute the data processing: it can *not* be held to cover "all the legitimate purposes" followed by the data controller [46]. Consent should refer to processing that is reasonable and necessary in relation to the purpose. However, the guidance is not clear on whether the processing operations must be granularly defined. WP 29 gives a negative example of how consent could fail to be specific: using data collected for movie recommendations to provide targeted advertisements. However, it does not provide a comprehensive set of tools for assessing specificity [47].

In its opinion on electronic health records (EHRs) [64], WP 29 provides that 'specific' consent must relate to a well-defined, concrete situation in which the processing of medical data is envisaged. Again, the meaning of 'concrete situation' is rather vague. According to the same opinion, if the reasons for which data are processed by the controller change at some point in time, the user must be notified and put in a position to consent to the new processing of personal data. The information supplied must discuss the repercussions of rejecting the proposed changes in particular.

One question that can be relevant at this point is whether any change in the purpose (no matter how insignificant) requires re-consent. Minor changes will probably not require new consent, while important ones will. But what is the yardstick to measure the discrepancy between the changes? Understanding the specific character of consent is essential for assessing the validity of consent expressed through the matching between preferences and requests. OAC functions based on subsumption between the requests for accessing personal data and the user preferences expressed in advance. Therefore, by hypothesis, preferences are expressed in broader terms, compared to requests, casting doubt on whether consent is specific. However, OAC also enables users to set prohibitions, and thus narrow down the scope of their consent. The reasoner that implements the matching transforms a preference, expressed through a positive statement (the purpose), and one or more negative ones (the prohibitions) into a choice between the available options as it materializes the preferences and produces legal effects towards third parties (the data controller).

Moreover, WP 29 also touches upon the relation between purposes and processing operations: "it should be sufficient in principle for data controllers to obtain consent only *once* for *different operations* if they fall within the *reasonable expectations* of the data subject" [46]. It is unclear, though, how these expectations are determined. They can be empirically identified, but the result would only be statistically relevant to the average data subject and not to the particular individual who consents. Another option would be to apply a framework for determining the contextual nature of the processing, such as the contextual integrity theory of privacy developed by Helen Nissenbaum [65]. In this theory, privacy is considered a right to the appropriate flow of personal information that identifies particular contexts and context-specific social norms. The context and the specific norms governing it can be useful to assess whether the requirement for consent to be specific is complied with.

### 4.2.2. Employing the Purpose Limitation Principle in Assessing the Specific Character of Consent

The purpose limitation principle in Article 5 (1) b) of the GDPR provides that personal data shall be "collected for specified, explicit and legitimate purposes and not further processed in a manner that is incompatible with those purposes". The principle can be separated into two components: (a) the purpose specification requirement and (b) the non-incompatibility requirement [66]. According to the WP 29 Opinion on Purpose Limitation [67], all the facts should be taken into account to determine the actual purpose, referring to the common understanding and reasonable expectation of the data subject, in the specific context of the case. Purpose specification is included in Article 8 of the Charter of Fundamental Rights of the European Union [68] concerning the right to protection of personal data: "Article (2) [...] data must be processed fairly for specified purposes and on the basis of the consent of the person concerned or some other legitimate basis laid down by law."

The purpose answers the question "why" regarding the processing of personal data and is different from the means of processing, which refer to "how" data are processed and from the interests relevant to the processing (which are formulated at a more abstract level) [66]. Similar to the requirement for consent to be specific, the requirement for a specific purpose is related to control, self-determination, and autonomy [66]. Compliance with a plethora of other principles and rules, including the principle of data minimization and storage limitation, can only be assessed in relation to a purpose [66]. The rationale for specific consent is to avoid the widening or blurring of purposes that may result in unanticipated use of personal data by the controller or by third parties and partially overlaps with the reasons behind the requirements of purpose specificity. This would result in a loss of user control [45]. User control is related to the preservation of autonomy, enabling the data subject to exercise agency regarding the processing of her/his personal data [48]. It is also a means of mitigating power and information asymmetries. In the

absence of a safeguard, after the moment of collection, the power balance tilts in favor of the data controller who is in 'possession' of the data and can use it for their own interest.

Furthermore, the EDPB discusses the specific character of consent in connection with function creep. According to Koops [69], the 'creep' element refers to the imperceptibility of the change. This deprives the data subject of the opportunity to contest the change and assess its consequences. Returning to OAC, the risk of function creep is mitigated because the data request specifically instantiates the purpose of processing and this information is available to the data subject in the Pod, enabling her/him to exercise her/his rights, such as the right to withdraw consent.

In what concerns the requirement for granularity, the system can include a taxonomy of domain-specific purposes, for example, purposes related to biomedical research, as we will detail in Section 5, enabling the interface designer to cluster them in categories with different levels of specificity, as required by the specific context.

### 4.2.3. Specific Data Controllers? Or Categories of Recipients?

With OAC [27], data subjects can express explicit authorization for specific data controllers that are known at the moment when preferences are set. However, in order to grant such explicit authorization in advance, information needs to be conveyed to the users when first wanting to use a Solid application or should be available somewhere for the Solid user to check, e.g., through metadata present on an app store. This is currently missing from Solid implementations—as is visible in Figure 2, which illustrates the current consent dialogue shown to Solid users when they want to use an app—the name and the ID of the app are shown, but nothing else, no contact details, no policies or links to policies defined somewhere else. Also, there is no established marketplace for Solid apps with information about the provider and/or developers of said apps.

In this context, an authorization can take different forms and degrees of specificity. One option is to require the data subject to identify the data controller by name and contact details and to grant them access to the personal data in the Pod. This case does not pose problems in terms of specificity. However, the downside of this option is that the data subject would have to approve each new data controller.

A second option consists of authorizing controllers based on relevant criteria, such as industry, country of incorporation, or sector. The data subject would set a list of preferences, but would not identify the entities by name and contact details. Instead, the data subject could set certain parameters regarding the conditions that the entities accessing their data must comply with. While the second option has the advantage of flexibility, there are questions as to whether it complies with the legal requirements for expressing valid consent.

**A. The moment when the identity of the controller is disclosed** The first challenge is represented by the moment when the data subject is informed about the identity of the data controllers. In the first example, the identity of the data subject is revealed in advance.

In the second example, the user sets the criteria that a requester should comply with and the identity of the data controllers becomes available at the moment when the reasoner decides that the requester complies with the criteria set by the data subject (industry, country of incorporation or sector). The identity and contact details are not explicitly acknowledged or approved by the data subject before access is granted, but they are available for consultation in the Pod. These are part of the dynamic elements that the data subject cannot set in advance with the OAC profile.

According to Recital 42 GDPR, for consent to be informed, the data subject "should be aware at least of the identity of the controller and the purposes of the processing for which the personal data are intended". The EDPB Guidelines 05/2020 on consent under Regulation 2016/679 [45] note: "in a case where the consent is to be relied upon by multiple (joint) controllers or if the data is to be transferred to or processed by other controllers who wish to rely on the original consent, *these organisations should all be named*." Similarly, the guidelines on transparency under Regulation 2016/679 [70] emphasize the importance of

the identity of the controller by stating that changes to a privacy statement/notice that *should always be communicated* to data subjects include inter alia: a change in processing purpose; a change to the *identity* of the controller; or a change as to how data subjects can exercise their rights in relation to the processing. As mentioned, the specific character of consent and the informed character of consent are closely related. It is therefore likely that consent will not be valid if the data subject is not aware of the identity of the entity that processes personal data when he/she consents.

As explained in Section 3.2, the data subjects must be informed at the time when the data are obtained from them or at the time of the first communication between the data controller and the data subject. This information is made available to the data subject before their data are disclosed to the app provider, but they are not in a position to make a choice. The decision is taken automatically by the matching algorithm, without the intervention of the data subject. This would most likely not comply with the requirements for expressing valid consent. However, it could function as an information mechanism when the processing relies on other grounds for lawfulness (such as contract, legitimate interest, or compliance with the law).

Recital 39 GDPR refers to the principle of lawfulness, fairness, and transparency and provides that transparency "concerns, in particular, information to the data subjects on the *identity of the controller* and the purposes of the processing and further information to ensure fair and transparent processing [...]". Transparency is also referred to in Recital 58 GDPR, providing that transparency is of particular relevance in situations where the proliferation of actors and the technological complexity of practice make it difficult for the data subject to know and understand whether, *by whom* and for what purpose, personal data relating to him or her are being collected. These texts suggest that the identity of the data controllers is an important element for compliance with information and transparency obligations.

**B. The difference between controllers and recipients of personal data** Articles 13 and 14 GDPR provide a list of elements that must be communicated to the data subject. Both articles separate the information on (i) "the identity and the contact details of the controller and, where applicable, of the controller's representative" (Article 13 (1) (a) and 14 (1) (a) GDPR) from the information on (ii) "the recipients or categories of recipients of the personal data, if any" (Article 13 (1) (e) and 14 (1) (e) GDPR). This suggests that, under certain conditions, the recipients can be identified by category and not by identification details. This poses the following legal questions: What is then the difference between data controllers and recipients, in particular in a decentralized setting? Are the requesters *controllers* or *recipients*? And what is the relevance of this distinction?

Pursuant to Article 4 (9) GDPR, "'recipient' means a natural or legal person [...] to which the personal data are disclosed, whether a third party or not". A third party is, according to Article 4 (10) GDPR, "a natural or legal person, [...] other than the data subject, controller, processor and persons who, under the direct authority of the controller or processor, are authorized to process personal data".

EDPB's guidelines on controller and processor [71] provide that, as opposed to the concepts of controller and processor, the Regulation does not lay down specific obligations or responsibilities for recipients and third parties. These can be said to be *relative concepts* in the sense that they describe a relation to a controller or processor from a specific perspective, e.g., a controller or processor discloses data to a recipient. The EDPB gives the example of a controller who sends data to another entity. Irrespective of its role as controller or processor, this party is considered a recipient.

The WP 29 Guidelines on transparency under Regulation 2016/679 [70] state that the actual (named) recipients of the personal data, or the *categories of recipients*, must be provided. To comply with the principle of fairness, controllers must provide information on the recipients that is the *most meaningful for data subjects*. This Guidance further clarifies this requirement and states that "in practice, this will generally be the *named recipients*, so that data subjects know exactly who has their personal data". However, WP 29 mentions the

possibility of also informing the data subjects on the categories of recipients, and requires their identification, as specifically as possible, by indicating the type of recipient, i.e., by reference to the activities it carries out, the industry, sector and sub-sector and the location of the recipients.

In one interpretation, the guidance might refer to data processors (entities that process personal data on behalf of the data controller), leaving out data controllers which, because of their importance in exercising data subject rights, must be identified specifically. Recital 39 GDPR refers to the rationale of transparent communications, stating that "Natural persons should be made aware of risks, rules, safeguards and rights in relation to the processing of personal data and *how to exercise their rights* in relation to such processing". The matching between criteria and actual requests enables individuals to exercise their rights. Even though the data subjects do not acknowledge/agree to the identity and contact details of the requesters, this information is available in the Pod, and they can use it in order to exercise their rights, such as the right of access (Article 15 GDPR), the right to rectification (Article 16 GDPR), the right to erasure (Article 17 GDPR) or the right to withdraw consent (Article 7 (3) GDPR).

The strict information requirements of identifying data controllers by name and contact details seem to be adapted to the current web, where entities in charge of centralized datastores collect the data and share it afterward with other entities (as recipients). However, in the structure intermediated by Solid, the sharing takes place between the data subject (enabled by the Solid Pod) and an undefined number of app providers, developers, or other users. This makes it very difficult to identify all these entities by name and contact details. Vogel discusses this problem in the context of the Data Governance Act and argues it is an impediment in providing intermediary services [72].

To conclude this section, it is likely that agreeing to the processing of personal data without identifying the providers of Solid services, specifically in terms of their identity and contact details, will not be regarded as valid consent under the GDPR. However, in a certain interpretations and based on WP 29 Guidelines, it might serve as an information mechanism that can enable compliance with Articles 13 and 14 GDPR. There is work underway by Esteves and Pandit [73] to have registries of entities recorded in Solid Pods, using the PLASMA (Policy LAnguage for Solid's Metadata-based Access control) vocabulary, which can include details regarding the identity and contact details of controllers and recipients. Listing 6 presents an illustrative example of a registry of entities.

**Listing 6.** Registry of entities using the PLASMA vocabulary.

```
PREFIX dcat: <https://www.w3.org/ns/dcat#>
PREFIX plasma: <https://w3id.org/plasma#>
PREFIX dcterms: <http://purl.org/dc/terms/>
PREFIX xsd: <http://www.w3.org/2001/XMLSchema#>
PREFIX dpv: <https://w3id.org/dpv#>
PREFIX odrl: <http://www.w3.org/ns/odrl/2/>
PREFIX ex: <https://example.com/>

ex:EntityRegistry a dcat:Catalog, plasma:UserRegistry ;
    dcterms:created "2023-07-01T13:21:18"^^xsd:dateTime ;
    dcterms:publisher <https://solidweb.me/besteves4/profile/card#me>
    dcat:dataset ex:AppProviderA .

ex:AppProviderA a plasma:AppProvider ;
    dpv:hasName "App provider A" ;
    dpv:hasContact "appproviderA@email.com" ;
    odrl:hasPolicy ex:PrivacyPolicyA .
```

### 4.3. Is Consent Necessary for Compatible Purposes?

This section discusses the validity of consent based on a compatible matching of purposes. For example, if a participant in research expresses her will to have her data processed for Alzheimer's disease, which is a type or subcategory of "degenerative disease", will this consent be valid for a request for using the personal data for a study on "dementia", also a category of degenerative diseases?

As detailed in Section 2.2, OAC's proposed matching algorithm is currently based on subsumption, i.e., if a user policy allows access for purpose A and a data request for purpose B, which is a subcategory of A, comes in, then the access should be permitted. The same is valid for the other matching operations that can be made, e.g., on processing operations or personal data categories. Continuing with the example presented above, this means that if the participant has a user policy that states that her data can be used for research on Alzheimer's disease and the request is for research on degenerative diseases, then access is not allowed as the purpose of the user policy is more specific than the purpose of the request. On the other hand, if the participant has a user policy that states that her data can be used for research on degenerative diseases and a request for research on Alzheimer's comes in, then access is permitted since the purpose of the request is more specific, i.e., is a subcategory, than the purpose of the user policy. Another advantage of OAC is the possibility to express prohibitions, a factor that assists with the specificity of consent, i.e., users can allow access to data for medical research and prohibit it for particular areas of medical research, e.g., genetic engineering research.

However, OAC does not consider yet the matching of "compatible purposes", e.g., if the participant has a user policy that states that her data can be used for research on Alzheimer's, and a request comes in to use the data for research on dementia, does it mean that the access should be allowed since, like Alzheimer's, dementia is a subcategory of a degenerative disease? The introduction of a "compatibility matching" algorithm to Solid would improve the model. However, it is necessary to ascertain the legal effects of this operation. A first question is how to determine if one purpose is compatible with another and if the user wishes to use such a compatibility model to deal with data requests. In order to enable this matching for the particular use case of biomedical research, the work of Pandit and Esteves [74] of having ODRL and DPV policies for health data-sharing can be reused, as it provides a taxonomy of health-related research purposes, connects to other ontologies with taxonomies of diseases and reuses the matching algorithm of OAC. However, for the reasoner to be able to decide on compatibility, this information should be embedded in the used taxonomies of purposes, for instance, by adding a triple statement expressing that `:purposeX :isCompatible :purposeY`.

In the GDPR, compatibility is discussed in relation to the second component of the purpose limitation principle in Article 5 (1) (b) GDPR—the non-incompatibility requirement—and the criteria for assessing it are mentioned in Article 6 (4) of the GDPR, as follows:

(a) "any link between the purposes for which the personal data have been collected and the purposes of the intended further processing;"
(b) "the context in which the personal data have been collected, in particular regarding the relationship between data subjects and the controller;"
(c) "the nature of the personal data, in particular, whether special categories of personal data are processed [. . . ];"
(d) "the possible consequences of the intended further processing for data subjects;"
(e) "the existence of appropriate safeguards, which may include encryption or pseudonymisation."

From a technical perspective, the first two criteria (a) the link between purposes and (b) the context of processing, might be determined by automated means, as described above. However, the third and fourth criteria involve an evaluation that connects the nature of data and the possible consequences of use to the data subjects. The last criterion—the existence of technical safeguards—can be partially verified automatically on the basis of certifications.

However, the appropriateness of the safeguards is an assessment that is difficult to automate, as it involves an assessment of the risks to the rights and interests of the data subjects.

A second question is how are the requirements of expressing consent influenced by the compatibility analysis. The non-compatibility requirement is a type of use limitation that prohibits the processing of personal data for purposes that are incompatible with the purpose at the time of the data collection. The requirement for compatibility and the requirement of an appropriate legal basis are cumulative conditions. Compatibility of purposes cannot compensate for a lack of ground for lawfulness. Therefore, there is a need to either ask for a renewed (downstream) consent or to identify an alternative legal basis under Article 6 (1) of the GDPR such as legitimate interest, the necessity to process the data for entering into or performing a contract, or compliance with a legal obligation. In the example above, if the data subject consented to the use of her data for Alzheimer's disease and the matching algorithm determines that the purposes are compatible, this would not suffice to grant access to the resources in the Pod. It is also necessary to express consent to the compatible purpose (research for dementia). In addition, as we will further develop in Section 5.1, if special categories of data are processed, it is necessary to also identify an exception under Article 9 of the GDPR.

Even though it cannot compensate for a lack of consent, the compatibility assessment is useful because it is a precondition for the use of data for new purposes based on other legal grounds.

*4.4. The Fine Line Dividing Expressing and Delegating Consent*

In this section, we discuss whether an OAC-based system allows data subjects to express or delegate consent, as well as who bears the liability in case of mistakes in decentralized data-sharing settings.

**A. Is consent delegated in Solid?** The matching algorithm in OAC transforms a preference regarding the processing of personal data (desired outcome for a specific privacy-related situation) into a decision (the choice in a specific privacy situation *among available options*) [75]. The focus of this paper [75] is on using these concepts in empirical studies about attitudes towards privacy, but the distinction can serve in the discussion about the matching process proposed in the context of Solid.

When setting their preferences using the OAC profile, the data subjects define certain parameters within which their personal data can be disclosed and used, and agree that the choice that produces practical effects is made by a sequence of actions performed according to the parameters, such as subsumption and exclusion. Therefore, consent is provided not only to the categories of data, the purposes, and the entities that act as data controllers but also to the mechanism by which these preferences are transformed into choices.

Discussing consent in the context of biobanking, Boers et al. argue that consent should focus on the governance structure of a biobank, rather than on the specific study details, referring to this type of consent as "consent for governance" [76]. Le Métayer and Monteleone analyze the idea of automated consent as a shift from *consent to the use of personal data* to *consent to the use of a privacy agent*, defined as "a dedicated software that would work as a surrogate and automatically manage consent on behalf of the data subject" [77]. Sheehan [78] discusses ethical consent and draws a difference between first-order and second-order decisions. Second-order decisions are fundamentally different from first-order decisions in that the details relevant to the decision-maker concern the decision-making process and not the subject matter of the choice. Sheehan provides the example of placing an order in a restaurant. He is imagining a scenario in which a few friends go for dinner and, before the waiter can take their order, one of them leaves for a few moments and asks one of the people at the table to order for him. When making decisions to delegate decision-making, the individual makes choices and gives preference to something that he values: the trust in his companions, their knowledge about his taste in food, the information that they hold about the approximate value that he wants to spend, etc.

Applying the distinction between first- and second-order decisions to OAC, the question is whether consent to the parameters and the matching algorithm is a first- or second-order choice. For example, if the data subject consents to research for the public interest, this notion is further interpreted by the matching algorithm (possibly enriched by an ontology) and the final choice of approving a request is granular and specific. However, the choice is not made directly by the data subject, but indirectly, by delegating it to an OAC-based system. An idea for improving OAC is, instead of adding more information to the ontology, to make the reasoner algorithm "learn" to make such inferences. However, in this case, Article 22 of the GDPR on automated decision-making might have to be considered.

**B. Who bears the liability in case of mistakes?** If consent is not expressed by the user, but rather delegated, it is necessary to inquire into the effect of the consent between the data subject, the agent, and the app provider.

Is the result of the matching sufficient proof that valid consent was obtained? It is arguable that the app provider cannot rely on the outcome of the matching algorithm to demonstrate that valid consent was expressed. From a private law perspective, this issue is discussed in the context of the agency/mandate agreement. According to the 'appearance principle', under certain conditions, the will expressed by the agent is binding upon the principle (the data subject). If the choices do not accurately reflect the will of the data subject, the matter will be resolved between the data subject and the agent [77]. This has the advantage of providing legal certainty in contract law. However, the European Data Protection Law is more protective towards the data subject. From the perspective of the GDPR, the app provider, in its role as data controller, bears the responsibility to ensure and to demonstrate that consent was validly obtained (Article 6 (1) a) and 7 (1) GDPR). In Solid, this means that the app provider will have to understand and document the matching process and not only the result. Thus, the matching algorithm would have to be transparent and show how the matching was performed, so that the app provider can assess whether consent was valid. However, presenting the privacy preferences of the data subject involves an invasion of the data subject's privacy. Having information about the user's preferences, the controller can submit targeted requests that match the said preferences [27].

The liability for mistakes has a different regime if the Pod provider is considered a separate data controller. In this interpretation, the data subject consents to the processing of personal data (storage and making it available to third parties under certain conditions) in relation to the Pod provider (Controller 1). Based on the instruction from the data subject, the Pod provider makes the data further available to the app provider (Controller 2). In this case, each data controller would have to ensure that the processing operation (storage, transfer, further use) is based on a valid ground for lawfulness [71]. The liability for a mistake in the matching process and for invalid consent could be shared between the two controllers, according to the agreement concluded between them. The GDPR does not require a legal form for this arrangement, but the EDPB recommends that this arrangement is made in a binding document such as a contract that is made available to the data subject [71].

## 5. The Special Case of Biomedical Research

As explained in the previous sections, the strict requirements for obtaining consent under the GDPR impose burdensome obligations on data subjects. A requirement for separate agreements for each app provider and for each specific purpose results in repeated requests for consent. In the biomedical context, the likelihood of individuals being involved in decision-making regarding their data might be lower compared to other sectors because the incentives for individuals to participate in biomedical research are different. The benefits derived from participation are usually not immediate and do not reflect on the personal situation of the individual. Therefore, the amount of effort that individuals are willing to invest (checking the Pod for new requests, reading the information notices, and approving/disapproving new access requests) might be lower, compared to, for example, information society services. There are several provisions that suggest a more flexible approach to the requirements related to consent or even to move away completely from consent.

Recital 33 GDPR suggests that broad consent is acceptable for research. Under certain conditions, data subjects can express consent to "certain areas of scientific research", if "recognised ethical standards for scientific research" are observed. However, this possibility is limited, as individuals shall have the opportunity to "give their consent only to certain areas of research or parts of research projects". The concepts of "areas of research" or "part of research projects" are domain-specific notions that are not defined in the GDPR, which is an omnibus regulation. The work of Pandit and Esteves on DUODRL (https://github.com/besteves4/duo-odrl-dpv/, accessed on 25 October 2023) [74], inspired by the work of the Global Alliance for Genomics and Health (https://www.ga4gh.org/, accessed on 25 October 2023) on the Data Use Ontology [79], can be reused by data subjects and data controllers to create policies for health data-sharing, as it provides a taxonomy of health-related research purposes, connects to other ontologies with taxonomies of diseases, and includes the concepts to model projects and duties to use data, e.g., the requirement to have ethical approval, the requirement for collaboration with the study's investigator(s), or the requirement to return the generated results of the study. Listing 7 provides an example of a user policy that states that the dataset https://example.com/Dataset can be used for the purpose of health, medical or biomedical research, identified with the `duodrl:HMB` concept, in the context of Project X, `ex:ProjectX`, provided that the user or app requesting access can provide proof of having ethical approval, identified with the `duodrl:ProvideEthicalApproval` concept.

**Listing 7.** An example ODRL offer policy generated by https://solidweb.me/arya/profile/card#me, stating that a dataset can be accessed for the purpose of health, medical or biomedical research in the context of Project X, provided that the entity requesting data provides documentation of ethical approval.

```
PREFIX odrl: <http://www.w3.org/ns/odrl/2/>
PREFIX dcterms: <http://purl.org/dc/terms/>
PREFIX duodrl: <https://w3id.org/duodrl#>
PREFIX xsd: <http://www.w3.org/2001/XMLSchema#>
PREFIX oac: <https://w3id.org/oac#>
PREFIX gist: <https://ontologies.semanticarts.com/gist/>
PREFIX ex: <https://example.com/>

<https://example.com/offerForBiomedicalResearch> a odrl:Offer ;
    dcterms:description "Request to use data for biomedical research (HMB) on Project X." ;
    dcterms:creator <https://solidweb.me/arya/profile/card#me> ;
    dcterms:issued "2023-06-01T18:15:56"^^xsd:dateTime ;
    odrl:uid <https://example.com/offerForBiomedicalResearch> ;
    odrl:profile oac: ;
    dcterms:source duodrl:DUO_0000006, duodrl:DUO_0000021, duodrl:DUO_0000027 ;
    odrl:permission [
        odrl:assigner <https://solidweb.me/arya/profile/card#me> ;
        odrl:action oac:Use ;
        odrl:target <https://example.com/Dataset> ;
        odrl:duty [
            odrl:action duodrl:ProvideEthicalApproval
        ] ;
        odrl:constraint [
            odrl:and ex:purposeConstraint, ex:projectConstraint
        ]
    ] .

ex:purposeConstraint a odrl:Constraint ;
    dcterms:title "Purpose for access is to conduct health, medical or biomedical (HMB) research." ;
    odrl:leftOperand oac:Purpose ;
    odrl:operator odrl:isA ;
    odrl:rightOperand duodrl:HMB .

ex:projectConstraint a odrl:Constraint ;
    dcterms:title "Data can be used in the context of Project X." ;
    odrl:leftOperand duodrl:Project ;
    odrl:operator odrl:eq ;
    odrl:rightOperand ex:ProjectX .

ex:ProjectX a gist:Project .
```

This provision of Recital 33 GDPR is only present in the preamble of the Regulation, which does not have binding force and is not mirrored in the text of the GDPR. Furthermore, it was interpreted narrowly by the EDPS in its preliminary opinion on Scientific Research (2020) [80]. The Supervisor states that Recital 33 does not take precedence over the provisions requiring consent to be specific, but also suggests an evaluation based on the rights of the data subject, the sensitivity of the data, the nature and purpose of the research, and the relevant ethical safeguards. At the same time, the EDPS mentions that, when purposes cannot be specified, data controllers could compensate by enhanced transparency and safeguards [80].

Looking beyond the European Union, the UK government proposed a prominent role for broad consent in medical research in its proposal for a reform of the Data Protection Act in the UK [81] and the proposal was well received, although some concerns were voiced regarding its lack of certainty and potential for abuse.

Moreover, the European Commission proposed a Regulation instrument governing the processing of health data, the European Health Data Space [82], which proposes to completely move away from consent for secondary use of personal data for biomedical research. This proposal defines a 'data holder' as "any natural or legal person, which is an entity or a body in the health or care sector, or performing research in relation to these sectors, as well as Union institutions, bodies, offices and agencies who has the right or obligation, in accordance with this Regulation, applicable Union law or national legislation implementing Union law, or in the case of non-personal data, through control of the technical design of a product and related services, the ability to make available, including to register, provide, restrict access or exchange certain data". In this case, data holders have an obligation to disclose personal and non-personal data, under certain conditions and for a restricted range of purposes, including scientific research (Article 34 (1) (e) [82]) without the consent of the data subject. Article 33(5) of the EHDS Proposal also seems to overrule national laws that require consent "where the consent of the natural person is required by national law, health data access bodies shall rely on the obligations laid down in this Chapter to provide access to electronic health data." It remains to be seen what the final versions of this proposal will be, whether consent will play a role, and whether it will be broad or specific. The proposal was criticized by the EDPB and EDPS in a joint opinion [83], which requires further clarity on the interplay between national laws requiring consent and the proposed European instrument.

In the GDPR, biomedical research poses challenges because it combines a stricter regime (because research involves processing health data, which are part of the GDPR's special categories of data) with a series of derogations (aiming at supporting research because of its importance for society).

### 5.1. A Stricter Regime

The processing of special categories of data (including data concerning health) is forbidden pursuant to Article 9 (1) GDPR. There are ten exceptions to this rule, one of which is the *explicit* consent of the data subject. However, as mentioned in the EDPB Opinion on Consent [45], it is unclear what the explicit character refers to, since expressing a "statement or clear affirmative action" is a prerequisite for "regular" consent. For the requirements for expressing consent in the GDPR, it needs to be clarified what extra efforts a controller should undertake in order to obtain the explicit consent of a data subject in line with the GDPR [45]. The EDPB provides several examples of expressing explicit consent. The Guidance [45] mentions a written statement or in the digital or online context: filling in an electronic form, sending an email, uploading a scanned document carrying the signature of the data subject, or using an electronic signature. Two-stage verification of consent is another option for expressing explicit consent. For example, a data subject may receive an email informing them of the controller's intention to process a medical data record. In the email, the controller states that he is requesting permission to use a specific collection of information for a specified reason. If the data subject agrees to the use of this information,

the controller requests an email response with the words 'I agree'. Following the receipt of the response, the data subject receives a verification link that must be clicked or an SMS message.

In Solid, this can be implemented in different ways. Depending on the Solid server where the users choose to host their Pod, an *inbox* container, similar to the email inboxes available in other ecosystems, might be present by default when the user creates the Pod and can be used to receive these special requests. This *inbox* container has a special access control authorization—other users, beyond the data subject of the Pod, can only write to the container, in order to ensure that only the data subject can read the resources in said container. However, since the presence of this container is not standardized across the Solid ecosystem, it cannot always be found or might be called something else, causing an interoperability problem, and hence applications cannot rely on its existence. A more refined solution relies on a graph-centric interpretation of a Pod, where *"each Solid pod is a hybrid, contextualized knowledge graph, wherein 'hybrid' indicates first-class support for both documents and RDF statements, and 'contextualized' the ability to associate each of its individual documents and statements with metadata such as policies, provenance, and trust"* [84]. With the proper recording of metadata, including context and provenance metadata, multiple views of the Pod can be generated as required by the different applications that the data subject wishes to use. In this case, the requests can be simply added to the graph, with no need to *hardcode* in the app where the requests should be written, and such requests can be visualized by the data subject using a Solid app or service compatible with this graphic-centric approach. Moreover, the work of Braun and Käfer [85] can be leveraged to sign and validate resources that carry the 'I agree' statement of the data subject.

We can conclude that it is difficult to express explicit consent by pre-set preferences. Matching user preferences (expressed in advance) with requests for processing personal data will, most likely, fail to comply with the explicit character of consent. While the matching can increase transparency and help the individual make a decision, a second action of approving the use of personal data is necessary in order to comply with the explicit character of consent.

### 5.2. A Series of Derogations

We are focusing on three aspects that are specific to the regime of health research: the presumption of compatibility between the purpose of collection and the use for research purposes, the limitation to the right to information, and alternative exceptions beyond consent that can be used to process special categories of data.

**A. The purpose limitation principle** The principle of purpose limitation and the compatibility assessment were discussed in Section 4.3 on "Is consent necessary for compatible purposes?". To remind the reader, pursuant to Article 5 (1) (b) of the GDPR, "data shall be collected for specific, explicit and legitimate purposes and not further processed in a manner that is incompatible with those purposes". When data are processed for scientific research purposes, there is a presumption of compatibility between the purpose of collection and further use, if the processing is conducted in accordance with appropriate safeguards for the rights and freedoms of the data subject (as provided under Article 89 (1) GDPR). It is important to emphasize that the prohibition to process personal data for incompatible purposes is different from the requirement of purpose specificity and a derogation does not reduce the requirement for a specific purpose. As previously explained in Section 4.3, irrespective of compatibility, the data controller would have to rely on consent or another legitimate basis laid down by law. However, one provision in the preamble of the GDPR questions the separation between the two requirements. Recital 50 GDPR reads as follows: "The processing of personal data for purposes other than those for which the personal data were initially collected should be allowed only where the processing is compatible with the purposes for which the personal data were initially collected. In such a case, no legal basis separate from that which allows the collection of the personal data is required." This seems

to challenge the separation between the purpose limitation principle and the principle of lawfulness. This intersection and the effect on Solid requires further legal research.

**B. The obligation to provide information** We discussed the information obligations in Articles 13 and 14 GDPR in Section 4.2.3, "Specific data controllers? Or categories of recipients?" focusing on the moment when the information needs to be provided to the data subject. If personal data are processed for (biomedical) research purposes, Article 14 GDPR provides an exception for cases when personal data have not been obtained from the data subject. This may apply to Solid if we consider that not all personal data stored in Solid Pods comes directly from the data subject, e.g., it can be generated by app providers, Pod providers, or other users or agents. Pursuant to Article 14 (5) GDPR if (i) the provision of information proves impossible or would involve a disproportionate effort or it is likely to render impossible or seriously impair the achievement of the objectives of the processing and (ii) the conditions and safeguards in Article 89 GDPR are respected, the general information obligations in Article 14 do not apply. Whether these conditions are complied with in Solid will have to be assessed on a case-by-case basis, depending on the context of the request for accessing the personal data in a Pod. However, it is likely that these conditions will be met in exceptional cases and not as a rule. In case the conditions are met, the data controller "shall take appropriate measures to protect the data subject's rights and freedoms and legitimate interests, including making the information publicly available". Further research is necessary to discuss the role of the notification system of Solid as an appropriate measure to protect the data subject's rights.

**C. Alternative legal bases beyond consent** Besides explicit consent, Article 9 (2) of the GDPR provides other exceptions from the prohibition to process special categories of data. Article 9 (2) (j) is especially relevant for this section because it refers to processing personal data for health research. This provision allows the processing of data concerning health when it is necessary for scientific research in accordance with Article 89(1) GDPR, based on Union or Member State law, which shall be proportionate to the aim pursued, respect the essence of the right to data protection and provide for suitable and specific measures to safeguard the fundamental rights and the interests of the data subject. Therefore, the application of this exception depends on the identification of a Union or Member State law that can serve as a basis for the processing of personal data. If the processing falls under the scope of such a law, the explicit consent of the data subject is not necessary.

What results from this section is that the derogations for processing personal data for scientific research depend on the implementation and application of appropriate safeguards. Pursuant to Article 89 (1) GDPR, these safeguards refer to respect for the principle of data minimization and consist of, for example, pseudonymization and techniques that do not permit the identification of data subjects. Future research can determine whether PIMS such as the matching system presented in this paper can play a role as a safeguard.

## 6. Concluding Remarks

The analysis presented in this paper shows that it is challenging to express consent in advance. We analyze consent as the process of setting preferences, including permissions (positive statements) and prohibitions (negative statements) expressed by the data subject and stored in Solid Pods. These statements are then matched with requests for processing personal data. Table 1 presents the main building blocks of expressing valid consent. Some of them were analyzed in this paper and others need to be further explored in future research. The validity of consent is heavily influenced by the interface built on top of Solid and the way in which the information is presented to and acknowledged by individuals.

Starting from our research question, how *Can the matching between user policies and data requests, in a decentralized setting such as the Solid project, signify lawful consent under the GDPR?*, we conclude that the GDPR does not oppose expressing consent in advance. However, we identify several requirements for obtaining valid consent and show the challenges of formalizing them in a decentralized setting. There is still much to be achieved when it comes to the alignment of Solid with legal requirements. In its current form, the

OAC system does not include all the necessary elements for expressing valid consent. The answer on whether consent is valid as a ground for lawfulness will depend on future technical developments, the analysis of several other legal elements, and on the design of the interfaces that will be developed on top of Solid.

**Table 1.** Overview of main building blocks for expressing consent and the discussion points raised in this paper.

| Building Block | GDPR | Level of Automation | Comment |
| --- | --- | --- | --- |
| Freely given | Article 4 (11) | The OAC profile allows users to express their preferences in terms of positive and negative statements. The free character of consent depends on the granularity of choices. The interface design will have an important role in determining whether consent is expressed freely. | This paper does not directly address the requirement of consent to be free, leaving out important discussions about imbalance of power, conditionality or detriment. However, it discusses one of the requirements of obtaining a free consent: granularity. |
| Specific | Article 4 (11) | It is possible to set specific preferences with the OAC profile (through positive and negative statements) by using semantic ontologies. However, in certain contexts, it might be necessary to obtain a new (downstream) consent that is more specific than the preferences expressed. Furthermore, we discuss whether it is necessary to also automate other elements, different from the purpose of processing, such as processing operations or recipients of the personal data. | The specific character of consent requires a contextual assessment. The current paper discusses how the specificity requirement relates to the purpose of processing, the processing operations and the identity of the data controller. We also raise the question of whether consent in OAC is expressed or delegated and the problems that this distinction raises. |
| Informed | Article 4 (11) | Some of the information requirements can be automated and included in the OAC profile. Further work is necessary to automate the requirements of Articles 13 and 14 GDPR. However, there are some dynamic elements (e.g., the identity of the recipients of personal data) that have to be communicated to the data subjects after they set their OAC preferences. The data subject would, in certain cases, need to consent to the processing of personal data considering these new elements. | In setting their privacy profile, individuals get informed about the future uses of their data. Furthermore, during the processing life cycle, new information becomes available in the Pod and can be accessed by the data subject. One challenge we discuss is the timing of the notification. While the Pod will contain the necessary details about the processing, the data subject is informed about them after the moment when she/he made a choice. From a legal perspective, it remains up to discussion what is the connection between the details concerning the processing mentioned in Articles 13 and 14 and the requirements for obtaining a valid consent. |

**Table 1.** *Cont.*

| Building Block | GDPR | Level of Automation | Comment |
|---|---|---|---|
| Unambiguous indication of [...] wishes | Article 4 (11) | The interface design will influence whether consent is expressed unambiguously. This requirement will be relevant at the moment when the user configures the OAC policies and at the moment when a downstream consent might be requested. | This requirement is not addressed in this paper. |
| By a clear affirmative action | When setting OAC policies, the user has to actively make choices and to express these choices. | This requirement is not addressed in this paper. | |
| The controller shall be able to demonstrate | Article 7 (1) | The tracking system of Solid/OAC can enable compliance with this requirement. | Although not a central point of the analysis, we touch upon the importance of Solid as a technical measure that can demonstrate consent. |
| (written declaration) [...] distinguishable from the other matters | Article 7 (2) | Consent should be separated from consent for other matters, both when the OAC preferences are set and when renewed consent is requested. | This requirement is not addressed in this paper and will depend on the design of the consent interface. |
| The right to withdraw consent | Article 7 (3) | The data subject has the right to withdraw consent at any time and without detriment and Solid must develop a system that enables the exercise of this right. | This requirement is not addressed in this paper. |
| In case of special categories of data: explicit consent for one or more specified purposes | Article 9 (2) (a) | This paper suggests solutions for expressing explicit consent. | Further research is needed on the implementation of these solutions. |

Going forward, we consider that Solid can serve an important role in ensuring compliance with data protection laws in two different ways: (a) aim for obtaining valid consent by including in the OAC profile more elements required by law and, if this is not possible, ask for new approval as they become known or better defined or (b) develop the OAC profile as a safeguard when other grounds for lawfulness, different from consent, are relied on, by ensuring transparency, traceability and user control.

As future work, we suggest several directions for further research: study the specificity of purposes and processing operations provided in taxonomies, such as the ones available in DPV, to check whether their labeling is enough for both data controllers to declare their activities and for data subjects to understand what is happening to their data, have tools to assess the compatibility of purposes to reduce the burden on users accessing similar data requests, develop a taxonomy of recipients, e.g., by industry, sector, etc., to express which recipient categories can, cannot or are receiving a copy of the personal data of the users, research on the value of PIMS for enabling compliance with legal requirements beyond consent and the use of OAC as an information mechanism and as a safeguard when other legal basis are used, implement a stricter access control mechanism for special categories of data (when health data are processed on the basis of Article 9 (2) (a) GDPR), for instance using Verifiable Credentials, and look at the requirements of new data-related laws being discussed and approved in the EU, such as the Data Governance Act, Data Act or the European Health Data Space proposal, such as the California Consumer Privacy

Act (CCPA), the General Personal Data Protection Law (or Lei Geral de Proteção de Dados Pessoais—LGPD) in Brazil or the Digital Personal Data Protection Act (DPDP) in India.

More specifically, future research should develop the current analysis regarding the informed character of consent. The current paper focuses on the information concerning the purpose of processing the identity of the data controller and the recipients or categories of recipients of the personal data (highlighted in Recital 42 GDPR). Future work is necessary to determine if the other requirements listed in Articles 13 and 14 of the GDPR, for example, details regarding the storage period, influence the validity of consent and whether they can be automated. It seems that not all elements are equally important for obtaining valid consent.

In what concerns the use of PIMS beyond consent (as a legal ground for lawfulness), future work should focus on exploring whether the OAC system can function as an information mechanism and as a safeguard, together with other grounds for lawfulness (such as Article 6 (1) (f) GDPR on legitimate interests or Article 6 (1) (c) GDPR on compliance with a legal obligation) and with the special conditions for processing health data (Article 9 (2) (j) GDPR on processing necessary for scientific research purposes).

**Author Contributions:** Conceptualization, M.F. and B.E.; methodology, M.F. and B.E.; validation, M.F. and B.E.; investigation, M.F. and B.E.; writing—original draft preparation, M.F. and B.E.; writing—review and editing, M.F. and B.E. All authors have read and agreed to the published version of the manuscript.

**Funding:** This article is partially funded by the COST Action on Distributed Knowledge Graphs (CA19134), supported by COST (European Cooperation in Science and Technology). Beatriz Esteves was funded by the European Union's Horizon 2020 research and innovation program under the Marie Skłodowska-Curie grant agreement No. 813497 (PROTECT). Marcu Florea is an Early Stage Researcher within the KnowGraphs Project, the work of which is supported by the European Union's Horizon 2020 Research and Innovation Program under the Marie Skłodowska-Curie Innovative Training Network, grant agreement No. 860801.

**Data Availability Statement:** Data are contained within the article.

**Acknowledgments:** The authors wish to thank Pat McBennett, Harshvardhan J. Pandit, Jeanne Mifsud Bonnici and Radina Stoykova for their valuable feedback that contributed to the improvement of this work.

**Conflicts of Interest:** The authors declare no conflict of interest.

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
