# Peer review of "Is Automated Consent in Solid GDPR-Compliant? An Approach for Obtaining Valid Consent with the Solid Protocol"

_information, doi:10.3390/info14120631_

Round 1

Reviewer 1 Report

Comments and Suggestions for Authors

Summary:

In this article the authors focus on the legal aspects and challenges of obtaining consent in Solid regarding GDPR. This also includes the discussion of the possibility to automate consent by preferences as well as technical challenges to achieve the aforementioned goals. The authors tackle an important research field by discussing the conditions for consent. 

General concept comments:

- The authors highlight several important problem statements regarding the usage conditions of personal data or the validity of consent, but some of the answers are left for future work. 
- The authors might also consider the impact of privacy policy languages on the requirement for transparency or the impact of pivacy preference languages on automated consent decisions. An overview can be found for example in the following paper: "Becher, Stefan & Gerl, Armin & Meier, Bianca & Bölz, Felix. (2020). Big Picture on Privacy Enhancing Technologies in e-Health: A Holistic Personal Privacy Workflow. Information. 11. 356. 10.3390/info11070356. "

Specific comments:

- line 25-26: "Data controllers (...) data must declare a lawful, fair and transparent purpose..."

Author Response

Firstly, we would like to thank the reviewer for the provided feedback for the improvement of the manuscript.

Regarding the provided comments, we added more references on the impact of privacy policy languages to the second paragraph of Section 2.2, including the paper suggested by the reviewer. Also, typos were found and fixed.

Reviewer 2 Report

Comments and Suggestions for Authors

The paper is interesting, it touches on important issues regarding data and health data (authentication and authorization).  The title is not appropriate and needs to be modified to a better new version, you may add for example Solid protocol at the end.  Figs. 2 and 3 need to be omitted or replaced by self-sounding clear figures.  Conclusion and future work need to be separated or be short in its current state in the paper. The paper is too long and there is a need to put some more diagrams and if possible some flowcharts 

Author Response

Firstly, we would like to thank the reviewer for the provided feedback for the improvement of the manuscript.

Regarding the provided comments, the title of the work was updated to “Is Automated Consent in Solid GDPR-Compliant? An Approach for Obtaining Valid Consent with the Solid Protocol”, in order to include references to GDPR compliance and to the Solid protocol. Figures 2 and 3 were improved with better quality images. In addition, as suggested by the reviewer, the last section was splitted into 2: Section 5 for ‘Future research directions’ and Section 6 for ‘Concluding remarks’. Also, some sections of the paper were better summarised to decrease its size and typos were found and fixed. The paper already includes three figures and several code examples to illustrate the work and, as such, we would like to hear from the reviewer more details on what specific diagrams and/or flowcharts would improve the manuscript.

Reviewer 3 Report

Comments and Suggestions for Authors

The paper focuses on Solid, which is a protocol for the decentralized storage of data, and poses the question of obtaining consent that aligns with the GDPR. The paper discusses the efforts to introduce policies in Solid for automating consent in advance and investigates this in the case of biomedical research.

The paper presents a detailed discussion on whether consent can be automated. However, it is not clear what the contribution of the paper is other than raising the question and discussing it. It will be useful to have a clear statement about the contribution of the paper.

The related work section seems to only focus on Solid. What about related research on other protocols?

The case for biomedical research is presented and discussed. However, it appears to be a discussion and not something that has been verified. Hence, it is unclear what value this brings other than being an interesting discussion.

In terms of the research question, at the end of the paper, there is still no concrete answer to the question. It would probably help to summarise the necessary considerations and the criteria for consent that are discussed in a table.

What are the limitations of this study?

Author Response

Firstly, we would like to thank the reviewer for the provided feedback for the improvement of the manuscript.

Regarding the provided comments, the main contributions of the paper were reinforced in the last paragraph of the introduction and are already described in the Concluding remarks section. More information on other similar initiatives was added on the 4th paragraph of the Introduction. The related work or background section focus on Solid as it was the chosen use case for this paper as it is a “free, community-led, developer-friendly, open-source initiative that delivers on the promise of decentralising the storage of data by relying on Web standards and on Semantic Web vocabularies to promote data and services interoperability”, as described in the same paragraph. Section 4.4, on the specific use case of health data and biomedical research, was improved with further references. Its inclusion is of particular interest as health data is a special category of data under the GDPR and as such different requirements apply to its processing, while there are also some exceptions for the usage of data for research.  A table that summarises the discussion points of the paper is available now in Section 5, where the limitations of the study are also discussed as future work that still needs to be performed.

Reviewer 4 Report

Comments and Suggestions for Authors

This paper analysed the current efforts to introduce a policy layer in the Solid ecosystem, in particular, related to the challenge of obtaining consent focusing on the GDPR. 

-- Can the solutions discussed be extended to other countries' privacy regulations? 

-- Who will monitor the policy layer? What if the entities misbehave? 

-- "... important to make a distinction between what can be enforced technologically and what can only be legally enforced ..." more details about the technical side could be added. 

Author Response

Firstly, we would like to thank the reviewer for the provided feedback for the improvement of the manuscript. The extension of the work to other privacy regulations was added as future work in Section 5 as it falls out of scope of the current paper to address other jurisdictional requirements in this particular work. However, since most data protection laws are GDPR-influenced, we don’t expect great difficulties in adding their requirements to this analysis. Section 4.2.3.2. (‘Who bears the liability in case of mistakes?’) was added to discuss accountability in case entities misbehave. More details on the technical side were added to Section 4 of the paper.

Round 2

Reviewer 3 Report

Comments and Suggestions for Authors

I thank the authors for revising the paper based on my comments. My comments have been addressed.

Author Response

We would like to thank the reviewer for the provided feedback for the improvement of the manuscript.